# A Genome-Wide Association Study of Protein, Oil, and Amino Acid Content in Wild Soybean (*Glycine soja*)

**DOI:** 10.3390/plants12081665

**Published:** 2023-04-16

**Authors:** Woon Ji Kim, Byeong Hee Kang, Sehee Kang, Seoyoung Shin, Sreeparna Chowdhury, Soon-Chun Jeong, Man-Soo Choi, Soo-Kwon Park, Jung-Kyung Moon, Jaihyunk Ryu, Bo-Keun Ha

**Affiliations:** 1Advanced Radiation Technology Institute, Korea Atomic Energy Research Institute, Jeongup 56212, Republic of Korea; wjkim0101@kaeri.re.kr (W.J.K.); jhryu@kaeri.re.kr (J.R.); 2Department of Applied Plant Science, Chonnam National University, Gwangju 61186, Republic of Korea; rkdqudgml555@naver.com (B.H.K.); wlsgml7026@naver.com (S.K.); shinsy011123@gmail.com (S.S.); sreeparna1996@gmail.com (S.C.); 3BK21 FOUR Center for IT-Bio Convergence System Agriculture, Chonnam National University, Gwangju 61186, Republic of Korea; 4Bio-Evaluation Center, Korea Research Institute of Bioscience and Biotechnology, Cheongju 28116, Republic of Korea; soonchunjeong@gmail.com; 5National Institute of Crop Science, RDA, Wanju 55365, Republic of Korea; mschoi73@korea.kr (M.-S.C.); sookwonpark@korea.kr (S.-K.P.); moonjk2@korea.kr (J.-K.M.)

**Keywords:** wild soybean, protein, oil, amino acid, single-nucleotide polymorphism, genome-wide association study, 180K Axiom^®^ Soya SNP array

## Abstract

Soybean (*Glycine max* L.) is a globally important source of plant proteins, oils, and amino acids for both humans and livestock. Wild soybean (*Glycine soja* Sieb. and Zucc.), the ancestor of cultivated soybean, could be a useful genetic source for increasing these components in soybean crops. In this study, 96,432 single-nucleotide polymorphisms (SNPs) across 203 wild soybean accessions from the 180K Axiom^®^ Soya SNP array were investigated using an association analysis. Protein and oil content exhibited a highly significant negative correlation, while the 17 amino acids exhibited a highly significant positive correlation with each other. A genome-wide association study (GWAS) was conducted on the protein, oil, and amino acid content using the 203 wild soybean accessions. A total of 44 significant SNPs were associated with protein, oil, and amino acid content. *Glyma.11g015500* and *Glyma.20g050300*, which contained SNPs detected from the GWAS, were selected as novel candidate genes for the protein and oil content, respectively. In addition, *Glyma.01g053200* and *Glyma.03g239700* were selected as novel candidate genes for nine of the amino acids (Ala, Asp, Glu, Gly, Leu, Lys, Pro, Ser, and Thr). The identification of the SNP markers related to protein, oil, and amino acid content reported in the present study is expected to help improve the quality of selective breeding programs for soybeans.

## 1. Introduction

Due to its high protein and oil content, soybean (*Glycine max* L.) is one of the world’s most important crops, accounting for the largest proportion of protein consumption, livestock feed, and oil seed production (http://soystats.com, accessed on 1 November 2022). Soybean protein contains all of the essential amino acids including isoleucine (Ile), histidine (His), leucine (Leu), lysine (Lys), methionine (Met), phenylalanine (Phe), threonine (Thr), tryptophan (Trp), and valine (Val), making it a nutritionally valuable crop [1]. In Korea, soybean is used in various products, including tofu, soymilk, soybean sprouts, and soybean paste. Therefore, the gene-based improvement of protein, oil, and amino acid content is a very important goal in soybean breeding. However, the domestication bottleneck and selective breeding have led to a significant reduction in the genetic diversity of modern soybean cultivars, which has hindered breeding progress [2].

Wild soybean (*Glycine soja* Sieb. and Zucc.), which is the ancestor of cultivated soybean (*G. max*), is highly valuable as a breeding material for the improvement of soybean because of its high genetic diversity [3,4]. The identification of genetic loci supporting the phenotypic diversity of protein, oil, and amino acid content observed in wild soybeans can be employed in soybean breeding. The average protein and oil content of cultivated soybean seeds is 40% and 20%, respectively [5], compared to 48% and 10% for wild soybean [6,7,8]. The amino acid composition of wild soybean is similar to that of cultivated soybean, with the highest content of glutamic acid (Glu) and aspartic acid (Asp) and the lowest content of cysteine (Cys) and Met [9,10].

To date, previous studies have reported 255 and 322 quantitative trait loci (QTLs) associated with the protein and oil content of soybean, respectively (https://www.soybase.org/, accessed on 1 November 2022). It has been widely documented that soybean protein and oil contents have a negative correlation [11,12]. Therefore, because a candidate gene for one of these traits may be associated with the other, they need to be studied together. Diers et al. (1992) identified major QTLs for protein and oil content on chromosomes 15 and 20 [13]; since then, these QTL regions have been gradually narrowed based on the results of previous studies [14,15,16,17,18]. These two major QTLs are found in the wild soybean accession PI 468916, with an increase in protein contents of 24 g/kg and 17 g/kg in the presence of homozygous alleles for the QTLs on chromosomes 20 and 15, respectively [13]. Recently, the QTL located on chromosome 15 was subjected to fine-mapping at 535 kb intervals between simple sequence repeat (SSR) markers [19], and fine-mapping and candidate gene selection have been completed for chromosome 20 as well [20].

Amino acids are widely used in the animal feed industry [1,21], with the poultry and swine industries consuming over 400,000 mt of Lys [1] and spending approximately USD 100 million annually to supplement feed with synthetic Met [22]. However, there have been few genetic studies targeting amino acid content compared to protein and oil contents in soybean. In a previous study, using a population of 101 F_6_-derived recombinant inbred lines (RILs) derived from the high-protein line N87-984-16 and the high-yield line TN93-99, a total of 32 QTLs associated with 18 amino acids were identified across 17 soybean chromosomes [23]. In particular, two QTLs associated with Cys were detected on molecular markers Satt235 (LG-G, chr. 18) and Satt252 (LG-F, chr. 13), and three QTLs associated with Met were detected on molecular markers Satt252 (LG-F, chr. 13), Satt564 (LG-G, chr. 18), and Satt590 (LG-M, chr. 7) [1]. Warrington et al. (2015) also identified a total of thirteen QTLs for the content ratio of four amino acids (Lys, Thr, Met, and Cys) in the crude protein from one hundred and forty RILs developed from a cross of Benning and Danbaekkong, and studied the relationship between the protein and amino acid contents [18]. Recently, eight genomic regions associated with the contents of Cys, Met, Lys, and Thr have been identified in a genome-wide association study (GWAS) using 621 *G. max* accessions in maturity groups I–IV and 34,014 single-nucleotide polymorphism (SNP) markers [21].

The objective of the present study was to identify candidate genes related to protein, oil, and amino acid content in a diverse set of 203 wild soybean accessions using a GWAS.

## 2. Results

### 2.1. Phenotypic Variation and Correlation Analysis

In 2015 and 2016, the seeds harvested from 203 wild soybeans were used to determine the protein, oil, and amino acid contents. The protein content was 38.61% to 52.61%, with an average of 47.88%, in 2015 and 38.83% to 53.21%, with an average of 47.92%, in 2016 (Table 1). The oil content ranged from 4.29% to 12.86% (an average of 7.11%) in 2015 and from 4.66% to 12.69% (an average of 7.51%) in 2016. The average coefficient of variation for the oil content was 18.5%, which was considerably higher than that for the protein content (5.1%). Based on the skewness and kurtosis of the data, the protein content exhibited a negatively skewed distribution, while the oil content was positively skewed (Table 1 and Figure 1). The broad-sense heritability (*h*^2^) for the protein and oil content was 0.78 and 0.83, respectively.

The contents of 17 amino acids, including Met and Cys, varied widely among the 203 wild soybean accessions (Table 2). Of these 17 amino acids, Glu was the main component of soybean seeds, with an average content of 7019 mg, while Cys had the lowest average content at 348 mg (Table 2 and Figure 2). On the other hand, Cys had the largest coefficient of variation with an average of 14%, but Thr had the lowest with an average of 4.9%. Arginine (Arg) had the highest *h*^2^ (0.71), while Met had the lowest (0.35).

The average correlations between the protein, oil, and amino acid contents in 2015 and 2016 are presented in Figure 3. The *p*-value of the correlation coefficients for each seed component was all significant at less than 0.001. The correlation between protein and oil content was negative at *r* = −0.64, while the correlation between each amino acid content was positive. Among the amino acids, Glu and Leu showed the highest correlation at *r* = 0.96, while Val and Met showed the lowest correlation at *r* = 0.34. In addition, each amino acid content had a positive correlation with protein content and a negative correlation with oil content.

### 2.2. Genome-Wide Association Study for Protein, Oil, and Amino Acid Content

A GWAS was conducted using a linear mixed model for the protein, oil, and 17 amino acid contents. Based on a significance threshold of −log_10_(*P*) ≥ 6.29, thirteen SNPs across six chromosomes and thirty SNPs across four chromosomes were detected for protein and oil, respectively (Figure 4). Of the detected SNPs, the most significant SNPs with the lowest *p*-value in a particular associated genetic region were selected as a causal SNP for the target trait. The selected SNP markers are summarized in Table 3. For the protein content, six SNP markers were identified on chromosomes three (AX-90486230), eleven (AX-90422214), twelve (AX-90436656), thirteen (AX-90336510), fourteen (AX-90450715), and fifteen (AX-90368184), while seven SNP markers were identified for the oil content on chromosomes twelve (AX-90408186 and AX-90513548), thirteen (AX-90440743), fourteen (AX-90525501), and twenty (AX-90387626, AX-90339137, and AX-90513791).

The allelic effect of the SNP marker was estimated for each trait, and the highest differences per trait are displayed in Figure 5. The protein-associated SNP marker AX-90422214 on chromosome 11 had the alleles G/A, and the average protein content for individuals with GG alleles was 42.84 g, which was 5.35 g lower than the average protein content for individuals with AA alleles (48.19 g). In addition, the oil-associated SNP marker AX-90513548 on chromosome 12 had the alleles C/T, and the average oil content for the individuals with CC alleles was 11.11 g, which was 3.98 g more than the average oil content for individuals with TT alleles (7.13 g).

At a suggestive threshold of −log_10_(*P*) ≥ 4.98, nineteen SNPs across five chromosomes were associated with Ala, six SNPs across three chromosomes with Arg, twenty SNPs across five chromosomes with Asp, two SNPs across two chromosomes with Cys, fourteen SNPs across three chromosomes with Glu, thirteen SNPs across five chromosomes with Gly, two SNPs across two chromosomes with His, fifteen SNPs across four chromosomes with Leu, forty-six SNPs across twelve chromosomes with Lys, three SNPs across two chromosomes with Phe, twenty-four SNPs across eight chromosomes with Pro, sixteen SNPs across seven chromosomes with Ser, twelve SNPs across four chromosomes with Thr, and three SNPs on one chromosome with Val (Appendix A and Table 4). Only one SNP was detected for each of Iso and Tyr, and no suggestive or significant SNPs were detected for Met. AX-90332294 on chromosome one and AX-90522787 on chromosome three were individually or simultaneously associated with eleven of the seventeen amino acids (all except Arg, Cys, His, Phe, Val, and Met). In addition, the AX-90397199 marker, which was located in almost the same position as the AX-90522787 marker on chromosome three, was associated with Leu and Lys.

The allelic effects of SNP markers on amino acid contents are presented in Figure 6. Glu, which exhibited the largest difference in the allele effects, was associated with SNP marker AX-90332294 on chromosome one, which had the alleles G/T, and the average Glu content for individuals with GG alleles was 8829 mg, which was 1033 mg higher than the average Glu content for individuals with TT alleles (7797 mg/g). On the other hand, Cys, which had the smallest allele effect, was associated with SNP marker AX-90358159 on chromosome seven, which had the alleles C/T, and the average Cys content for individuals with CC alleles was 540 mg, which was 104 mg higher than the average Glu content for individuals with TT alleles (437 mg/g).

### 2.3. Candidate Genes for Trait-Associated SNP Markers

Candidate genes for selected SNPs from the GWAS results were searched for approximately 50 kb upstream and downstream of the SNP. Protein- and oil-associated SNP markers were located in four (*Glyma.11g015500*, *Glyma.12g063800*, *Glyma.13g128700*, and *Glyma.15g055200*) and six (*Glyma.12g183500*, *Glyma.13g099400*, *Glyma.14g089900*, *Glyma.20g032400*, *Glyma.20g050300*, and *Glyma.20g087700*) genes, respectively (Table 5).

On chromosome one (*Glyma.01g053100*, *Glyma.01g053200*, and *Glyma.01g053900*) and chromosome three (*Glyma.03g239700*, *Glyma.03g240100*, and *Glyma.03g242400*), three candidate genes each were detected for nine amino acids (Ala, Asp, Glu, Gly, Leu, Lys, Pro, Ser, and Thr). The candidate genes detected for Arg and His were *Glyma.11g014900* and *Glyma.11g015500* on chromosome 11, while the candidate gene detected for Arg, Lys, and Phe was *Glyma.20g025400* on chromosome 20. In addition, *Glyma.06g061700* and *Glyma.06g064700* on chromosome six were detected for Val, *Glyma.12g072500* on chromosome twelve for Ala, *Glyma.13g099400* on chromosome thirteen for Lys, *Glyma.15g012300* and *Glyma.15g015800* on chromosome fifteen for Ala, and *Glyma.19g164800* and *Glyma.19g164900* on chromosome nineteen for Pro (Table 6).

## 3. Discussion

The wild soybean accessions used in this study were collected from Korea, China, Japan, and Russia and contain various genetic diversity [24], which can be utilized for soybean improvement by applying the GWAS to identify useful alleles. Soybeans contain not only essential amino acids but also a large amount of unsaturated fatty acids; thus, they are widely consumed for health purposes. As a result, many studies have been conducted on QTLs involved in regulating protein and oil content. The present study analyzed 203 wild soybean accessions grown for two years for their content of protein, oil, and 17 amino acids. The average protein and oil content for wild soybean was 47.84% and 7.33%, respectively. This is consistent with several previous studies that have reported a higher protein content and lower fat content than the 40% and 20% widely reported for protein and oil, respectively, in cultivated soybean [6,7,8]. Globally, soybean accounts for the largest proportion of oilseed production at 61% (http://soystats.com/, accessed on 1 November 2022). These results indicate that the oil content has increased during the domestication of cultivated soybean from wild soybean. On the other hand, there has been a shift toward lower protein content. This can be explained by the negative correlation between protein and oil content [11,12,21], which is also observed in Figure 3. In soybeans, there are constituent amino acids that make up the proteins, and there are free amino acids. In this study, the constituent amino acids were analyzed, and it was found that the content of Glu was the highest, and the correlation between each amino acid was significantly positive. Chotekajorn et al. (2021) [25] analyzed the free amino acids from 316 wild soybean accessions and identified that Arg was the most abundant, while most of the amino acids were positively correlated with each other, similar to the results of this study.

In the GWAS results, five and six genes containing detected SNP markers were found for the protein and oil content, respectively. It has been widely reported by many studies that major candidate genes associated with protein contents are present on chromosomes 15 and 20 [13,15,17,19,20]. Kim et al. (2016) conducted fine-mapping of the protein and oil content using a backcross population with the high-protein line PI 407788A as the donor parent and Williams 82 as the recurrent parent and found that QTLs were located between BARCSOYSSR_15_0161 and BARCSOYSSR_15_0194 on chromosome 15 [19]. In addition, Fliege et al. (2022) recently conducted fine-mapping and the RNAi transformation of the protein content using a backcross population with the high-protein wild soybean line PI 468916 as the donor parent and A81-356022 as the recurrent parent in the initial stages of a large-scale QTL analysis for soy protein and oil content [20]. Their research revealed that the protein content is regulated by a CCT domain protein polymorphism in the *Glyma.20G85100* gene on chromosome 20 [20]. In this study, AX-90368184 on chromosome 15 and AX-90513791 on chromosome 20, which were associated with protein and oil content, respectively, were detected at positions similar to the aforementioned major QTLs. The genes on the reference genome where the SNPs are located are *Glyma.15g055200* (F-box and associated interaction domain-containing protein) and *Glyma.20g087700* (protein of unknown function), which differ from the aforementioned genes. However, it is clear that the major QTLs are located on chromosomes 15 and 20.

These differences can be ascribed to the analysis of different accessions and the fact that the protein and oil content is not regulated by a single gene. The DNA binding with one finger (DOF) family of plant-specific transcription factors (TFs) is known to regulate seed protein accumulation and mobilization [26]. OBP3, an annotation of the *Glyma.11g015500* gene detected on chromosome 12, is a member of the DOF family. It has been reported that OBP3 regulates the signaling of phytochrome and tryptochrome in Arabidopsis thaliana [27] and plays an important role in growth and development [26]. However, the function of OBP3 in soybean is unknown. The involvement of the DOF family in protein accumulation suggests that the *Glyma.11g015500* gene may be a strong candidate gene for involvement in regulating the protein content.

The *Glyma.20g050300* gene for zinc-binding alcohol dehydrogenase family protein was detected on chromosome 20 and associated with oil contents. Soybean alcohol dehydrogenase has been found to be active in anaerobic reactions and seed respiration, including in response to flooding stress [28,29]. On the other hand, alcohol dehydrogenase was included among the fatty acid synthesis-related proteins identified in the comparative proteomics of high-fat soybean cultivar JY73 in a previous study [30]. Therefore, the fat content may be indirectly affected by alcohol dehydrogenase depending on the condition of the seed; thus, *Glyma.20g050300* may be a candidate gene for the regulation of the oil content.

Interestingly, in the GWAS results for the content of the seventeen amino acids, markers AX-90332294 and AX-90522787 located on chromosomes one and three, respectively, were detected for nine amino acids. In particular, AX-90332294 exhibited the largest SNP variance for each amino acid (Figure 6). For these amino acids, *Glyma.01g053200* and *Glyma.03g239700*, which were annotated with the prefoldin chaperone subunit family protein and aspartyl protease/7S seed globulin precursor, respectively, were identified. Chaperone is known to act as a proteolytic enzyme in eukaryotes by inducing proteases that aid in the structural folding of protein complexes or the degradation of proteins [31,32]. Prefoldins are a family of chaperone proteins, which are heterohexameric proteins composed of two α subunits and four β subunits [31,33]. Protein complexes are eventually formed by amino acids, so *Glyma.01g053200* may be related to the content of amino acids. In addition, it is known that β-con-glycinin (7S) and glycinin (11S) account for more than 70% of the total soybean storage proteins [23,34]. The fact that the 7S and 11S proteins in soybeans make up a significant portion of storage proteins may not be directly related to the presence of SNPs in structural genes. Rather, the expression and accumulation of these proteins are known to be controlled by regulatory elements, such as promoters and enhancers, that govern the transcription and translation of the corresponding genes. However, genetic variation, including SNPs in structural genes encoding 7S and 11S proteins, can affect expression levels or protein function, which in turn can affect soybean protein composition and nutritional value. Thus, while the presence of SNPs in structural genes may not be directly related to the abundance of storage proteins in soybeans, in a broader sense, it suggests that genetic variation in these genes may have important implications for the quality and utilization of soybean proteins. In this study, *Glyma.03g239700*, *Glyma.19g164800*, and *Glyma.19g164900* are associated with precursors or subunits of 7S and 11S storage proteins. In addition, amino acid synthesis involves several complex processes [35] and can be regulated by shikimate dehydrogenase (*Glyma.03g242400*), chorismate mutase (*Glyma.06g061700*), arogenate dehydratase (*Glyma.12g072500*), asparagine (*Glyma.20g025400*), and aminotransferase (*Glyma.15g012300*). Gene expression patterns for candidate genes from https://www.soybase.org/soyseq/, accessed on 1 November 2022 are shown in Appendix A. Candidate genes were expressed according to various tissues and stages of seed development, and in particular, it was confirmed that *Glyma.03g239700* was intensively expressed during the period of seed development. This information suggests that the candidate genes detected in the present study may directly or indirectly affect amino acid contents.

## 4. Materials and Methods

### 4.1. Plant Materials and Field Management

A total of 203 wild soybean accessions were used for analysis [24]. Briefly, they were cultivated and harvested in an experimental field at Chonnam National University (Gwangju, 36°17′ N, 126°39′ E, Republic of Korea) in 2015 and 2016. Each accession was planted in a single hill plot measuring 1 × 1 m in two replicates in the experimental field. A compound fertilizer with a ratio of nitrogen, phosphorus pentoxide, and potassium oxide of 8:8:9 was applied at 40 kg per 1000 m^2^.

### 4.2. Analysis of Protein, Oil, and Amino Acid Content

Seed samples were dried in a dry oven at 40 °C for 7 days, and then finely ground and quantified by 3 g each. The protein and oil contents were measured using the Kjeldal method and the ether extraction method, respectively, in the same way as used by Kim et al. (2023) [36]. The composition and content of the amino acids in the soybean seeds were analyzed using an amino acid analyzer (S433-H, SYKAM GmbH, Munich, Germany). Briefly, in the pretreatment process, 0.1 g of the sample was weighed into an 18 ml test tube, and 5 mL of 6N hydrochloric acid (HCl) was added. The tube was sealed under reduced pressure (nitrogen gas filling) and then hydrolyzed in a heating block set at 110 °C for 24 h. After hydrolysis was completed, the acid was removed via rotary evaporation at 50 °C. Using 10 mL of a sodium dilution buffer (pH 3.45–10.85), 1 mL of the sample was then filtered through a 0.2 μm membrane filter in a cation separation column (LCA K06/Na, 4.6 × 150 mm), with a flow rate for the buffer solution of 0.45 mL/min, a flow rate for the reagent of 0.25 mL/min, and a column temperature of 57–74 °C. The amino acids were identified using a fluorescence spectrophotometer at wavelengths of 440 and 570 nm.

### 4.3. DNA Extraction and SNP Genotyping

Leaf tissue was collected from young trifoliate leaves of the V2 seedling stage and ground to a fine powder using liquid nitrogen in a mortar. The genomic DNA extraction of the ground leaf tissue was carried out according to the manufacturer’s instructions using a DNeasy Plant Mini Kit (QIAGEN, Valencia, CA). The quantity and quality of the extracted total DNA were testified using a Nano-MD UV-Vis spectrophotometer (Scinco, Seoul, Republic of Korea). A total of 203 wild soybean accession genotyping was performed using the 180K Axiom Soya SNP array (Affymetrix, CA, USA) [37]. Low-quality SNPs were eliminated by removing SNPs when a genotype was observed in less than 95% of the samples and SNPs with a minor allele frequency (MAF) of less than 5%, and duplicates in the raw data were removed using R software [38]. Missing genotypes were estimated using BEAGLE software 3.0 [39], resulting in a total of 96,432 SNPs used in the analysis.

### 4.4. Genome-Wide Association Study and Statistical Analysis

The method of the GWAS was the same as Kim et al. (2023) [24]. Briefly, a linear mixed model using the restricted maximum likelihood (REML) algorithm was used in consideration of the population structure and similarity matrix. In addition, association analysis was performed using merged phenotypes to increase resolution between environments, and −log_10_(*P*) thresholds in Manhattan plots were calculated using the Bonferroni method (*P* = α/n). With 96,432 SNPs used in this study, at α = 1 and α = 0.05, the Bonferroni-corrected thresholds for the *p*-values were 1.04 × 10^−5^ (α = 1) and 5.19 × 10^−7^ (α = 0.05), with equivalent −log_10_(*P*) values of 4.98 for the suggestive threshold and 6.29 for the significance threshold [40]. The soybean reference genome, Glyma.Wm82.a2.v1, from https://www.soybase.org (accessed on 1 November 2022), was used as the gene model for candidate gene identification. Phenotypic data for each trait were subjected to descriptive statistics and correlation analysis using Microsoft Excel 2016. Broad-sense heritability (*h*^2^) and the GWAS analysis were calculated using QTLmax V2 software [41], and the following formula was used to calculate *h*^2^: $$h^{2}=\frac{\sigma_{u}^{2}}{\sigma_{u}^{2}+\sigma_{E}^{2}}$$

## 5. Conclusions

In this study, a GWAS analysis was conducted for protein, oil, and amino acid content using 203 wild soybean accessions. SNP markers for protein and oil content were detected at novel loci along with loci on chromosomes 15 and 20, which have been reported as major locations by several previous studies. In addition, candidate genes *Glyma.01g053200* and *Glyma.03g239700* were also found to have the potential to affect the content of nine amino acids. These findings could thus prove useful for soybean breeding programs.

## Figures and Tables

**Figure 1 plants-12-01665-f001:**
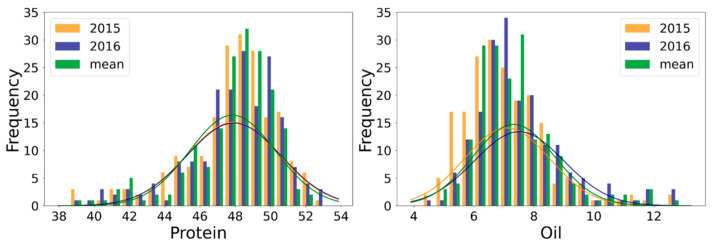
Distribution of protein and oil content in 203 wild soybean accessions.

**Figure 2 plants-12-01665-f002:**
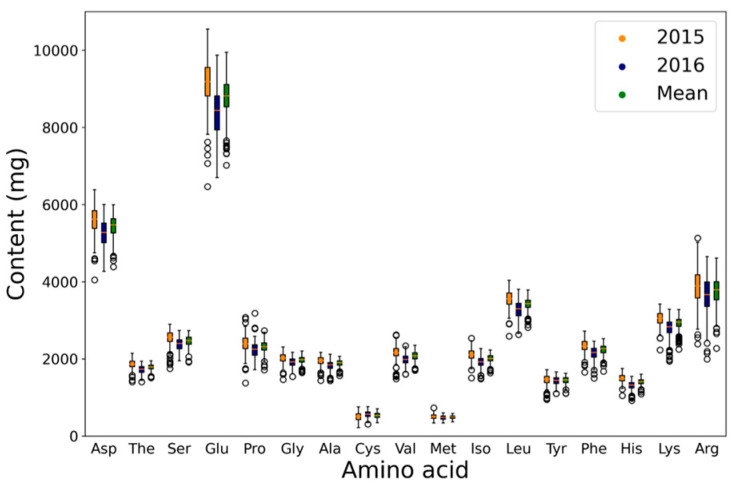
Distribution of protein and oil content in 203 wild soybean accessions. Asp, aspartic acid; Thr, threonine; Ser, serine; Glu, glutamic acid; Pro, proline; Gly, glycine; Ala, alanine; Cys, cysteine; Val, valine; Met, methionine; Iso, isoleucine; Leu, leucine; Tyr, tyrosine; Phe, phenylalanine; His, histidine; Lys, lysine; Arg, arginine.

**Figure 3 plants-12-01665-f003:**
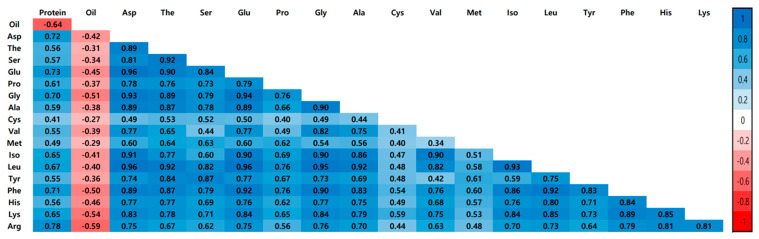
Correlation analysis among the seed component of 203 wild soybean accessions for the average of 2015 and 2016. Asp, aspartic acid; Thr, threonine; Ser, serine; Glu, glutamic acid; Pro, proline; Gly, glycine; Ala, alanine; Cys, cysteine; Val, valine; Met, methionine; Iso, isoleucine; Leu, leucine; Tyr, tyrosine; Phe, phenylalanine; His, histidine; Lys, lysine; Arg, arginine. All components of each have a *p*-value of <0.001.

**Figure 4 plants-12-01665-f004:**
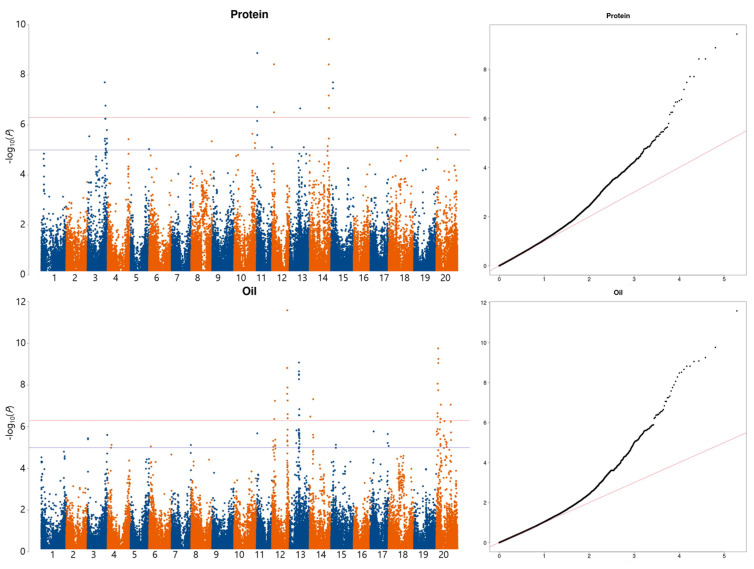
Manhattan plots and quantile–quantile (QQ) plots for protein and oil content in 203 wild soybean accessions. In the Manhattan plots, the blue line indicates the genome-wide threshold −log_10_(*P*) = 4.98 and the red line represents −log_10_(*P*) = 6.29, calculated using the Bonferroni method.

**Figure 5 plants-12-01665-f005:**
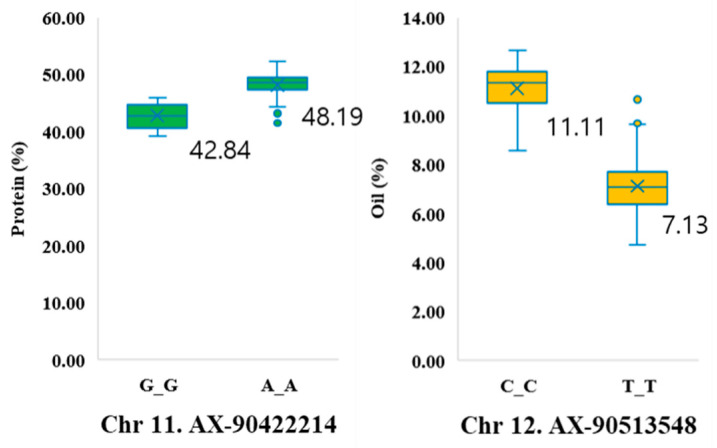
Phenotypic differences between lines carrying different SNP alleles associated with protein and oil content.

**Figure 6 plants-12-01665-f006:**
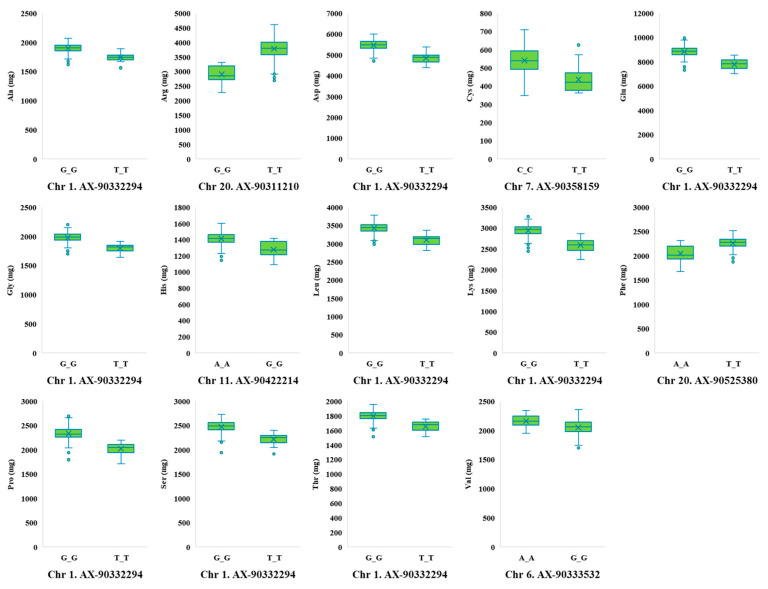
Phenotypic differences between lines carrying different SNP alleles associated with amino acid contents. Asp, aspartic acid; Thr, threonine; Ser, serine; Glu, glutamic acid; Pro, proline; Gly, glycine; Ala, alanine; Cys, cysteine; Val, valine; Leu, leucine; Phe, phenylalanine; His, histidine; Lys, lysine; Arg, arginine.

**Table 1 plants-12-01665-t001:** Descriptive statistics for the protein and oil content in 2015 and 2016.

Trait	Env.	Min.	Max.	SD	Mean	CV (%)	Skew	Kur.	*h* ^2^
Protein (%)	2015	38.61	52.61	2.64	47.88	5.5	−1.05	1.16	0.78
	2016	38.83	53.21	2.66	47.92	5.6	−1.01	1.20	
	Mean	39.17	52.27	2.43	47.84	5.1	−1.26	1.33	
Oil (%)	2015	4.29	12.86	1.42	7.11	19.9	1.16	2.35	0.83
	2016	4.66	12.69	1.49	7.51	19.8	1.20	1.88	
	Mean	4.74	12.68	1.36	7.33	18.5	1.33	2.52	

Env., environment; Min., minimum; Max., maximum; SD, standard deviation; CV, coefficient of variation; Skew, skewness; Kur., kurtosis; *h*^2^, broad sense heritability.

**Table 2 plants-12-01665-t002:** Descriptive statistics for the amino acid contents in 2015 and 2016.

Trait	Env.	Min.	Max.	SD	Mean	CV (%)	Skew	Kur.	*h* ^2^
Asp	2015	4047	6381	367	5592	6.6	−0.65	1.19	0.59
(mg/g)	2016	4265	6001	371	5242	7.1	−0.50	−0.31	
	Mean	4386	5994	304	5417	5.6	−0.79	0.51	
Thr	2015	1406	2142	114	1859	6.1	−0.68	1.54	0.46
	2016	1404	1939	110	1716	6.4	−0.53	−0.23	
	Mean	1512	1954	87	1788	4.9	−0.54	0.61	
Ser	2015	1751	2898	208	2531	8.2	−1.17	1.69	0.47
	2016	1950	2736	170	2377	7.2	−0.36	−0.44	
	Mean	1915	2733	148	2454	6.0	−0.76	0.69	
Glu	2015	6462	10,548	641	9138	7.0	−0.71	1.37	0.55
	2016	6694	9872	644	8373	7.7	−0.46	−0.30	
	Mean	7019	9947	521	8754	5.9	−0.82	0.69	
Pro	2015	1372	3075	222	2390	9.3	−0.60	2.71	0.45
	2016	1718	3184	201	2237	9.0	0.31	2.08	
	Mean	1714	2730	164	2314	7.1	−0.52	1.05	
Gly	2015	1462	2311	124	2021	6.2	−0.86	1.92	0.53
	2016	1541	2171	119	1908	6.2	−0.53	−0.06	
	Mean	1644	2200	99	1965	5.0	−0.75	0.45	
Ala	2015	1443	2169	120	1946	6.2	−0.91	1.32	0.44
	2016	1437	2120	125	1826	6.8	−0.80	0.74	
	Mean	1564	2066	95	1887	5.0	−0.81	0.54	
Cys	2015	223	757	102	503	20.2	−0.33	−0.35	0.47
	2016	310	764	84	558	15.0	−0.36	−0.22	
	Mean	348	709	74	531	14.0	−0.19	−0.34	
Val	2015	1480	2625	188	2163	8.7	−0.74	2.37	0.46
	2016	1599	2338	143	1980	7.2	−0.24	0.02	
	Mean	1700	2354	132	2071	6.4	−0.50	0.24	
Met	2015	339	735	62	499	12.5	0.31	0.35	0.35
	2016	341	602	56	478	11.8	−0.02	−0.51	
	Mean	369	584	44	489	9.0	−0.05	−0.38	
Iso	2015	1505	2536	148	2107	7.0	−0.14	1.07	0.53
	2016	1480	2269	143	1904	7.5	−0.51	0.17	
	Mean	1638	2231	117	2006	5.8	−0.56	0.13	
Leu	2015	2587	4035	220	3546	6.2	−0.69	1.43	0.50
	2016	2632	3802	228	3269	7.0	−0.43	−0.25	
	Mean	2817	3784	177	3407	5.2	−0.78	0.55	
Tyr	2015	947	1719	137	1460	9.4	−1.01	1.85	0.48
	2016	1101	1660	109	1429	7.6	−0.69	0.16	
	Mean	1106	1627	98	1445	6.8	−0.91	1.11	
Phe	2015	1657	2722	163	2338	7.0	−0.65	1.36	0.60
	2016	1501	2459	181	2140	8.5	−0.85	0.72	
	Mean	1677	2522	142	2238	6.3	−0.97	1.12	
His	2015	1048	1752	108	1496	7.2	−0.24	0.96	0.54
	2016	922	1545	114	1308	8.7	−0.81	0.82	
	Mean	1090	1603	90	1403	6.4	−0.68	0.66	
Lys	2015	2235	3420	186	3048	6.1	−0.64	1.15	0.50
	2016	1943	3285	254	2777	9.2	−1.12	1.20	
	Mean	2249	3279	177	2913	6.1	−1.04	1.07	
Arg	2015	2387	5128	451	3873	11.6	−0.28	0.52	0.71
	2016	1993	4648	486	3622	13.4	−0.64	0.54	
	Mean	2273	4613	411	3747	11.0	−0.63	0.60	

Asp, aspartic acid; Thr, threonine; Ser, serine; Glu, glutamic acid; Pro, proline; Gly, glycine; Ala, alanine; Cys, cysteine; Val, valine; Met, methionine; Iso, isoleucine; Leu, leucine; Tyr, tyrosine; Phe, phenylalanine; His, histidine; Lys, lysine; Arg, arginine; Env., environment; Min., minimum; Max., maximum; SD, standard deviation; CV, coefficient of variation; Skew, skewness; Kur., kurtosis; *h*^2^, broad heritability.

**Table 3 plants-12-01665-t003:** Significant SNP markers related to protein and oil content in the merged phenotype.

Trait	SNP	Chr	Position	−log_10_(*P*)	Reference ^+^	Minor	Major	MAF
Protein	AX-90486230	3	39,216,582	7.70	C	C	T	0.05
	AX-90422214	11	1,067,467	8.87	G	G	A	0.06
	AX-90436656	12	4,681,230	8.42	T	T	A	0.05
	AX-90336510	13	24,160,907	6.66	C	C	T	0.05
	AX-90450715	14	44,061,946	9.43	T	T	G	0.06
	AX-90368184	15	4,332,019	7.70	T	T	A	0.07
Oil	AX-90408186	12	6,713,247	7.24	A	A	C	0.06
	AX-90513548	12	34,460,746	8.82	C	C	T	0.06
	AX-90440743	13	21,445,098	9.08	C	C	T	0.08
	AX-90525501	14	8,179,883	7.32	T	T	A	0.08
	AX-90387626	20	4,185,011	9.76	C	C	G	0.06
	AX-90339137	20	10,234,522	7.06	A	A	C	0.05
	AX-90513791	20	32,766,317	7.06	T	T	C	0.05

SNP, single-nucleotide polymorphism; Chr, chromosome; Minor, minor allele; Major, major allele; MAF, minor allele frequency; + gene model: Glyma.Wm82.a2.v1.

**Table 4 plants-12-01665-t004:** Significant SNP markers related to amino acid contents in the merged phenotype.

Trait	SNP	Chr	Position	−log_10_(*P*)	Reference ^+^	Minor	Major	MAF
Ala	AX-90332294	1	6,633,674	5.19	G	T	G	0.07
	AX-90522787	3	43,983,721	6.73	G	G	T	0.08
	AX-90390655	12	5,348,412	6.39	G	G	T	0.16
	AX-90434969	15	1,252,401	5.87	A	A	G	0.15
Arg	AX-90392478	7	36,935,345	5.15	T	C	T	0.10
	AX-90422214	11	1,067,467	6.20	G	G	A	0.06
	AX-90311210	20	2,947,656	5.42	G	G	T	0.05
Asp	AX-90332294	1	6,633,674	6.03	G	T	G	0.07
	AX-90522787	3	43,983,721	6.77	G	G	T	0.08
Cys	AX-90358159	7	18,191,224	5.12	T	T	C	0.10
Glu	AX-90332294	1	6,633,674	6.16	G	T	G	0.07
	AX-90522787	3	43,983,721	6.67	G	G	T	0.08
Gly	AX-90332294	1	6,633,674	5.50	G	T	G	0.07
	AX-90522787	3	43,983,721	6.01	G	G	T	0.08
His	AX-90422214	11	1,067,467	5.10	G	G	A	0.06
Leu	AX-90332294	1	6,633,674	6.41	G	T	G	0.07
	AX-90397199	3	43,985,440	6.67	C	C	T	0.08
Lys	AX-90332294	1	6,633,674	6.02	G	T	G	0.07
	AX-90485849	1	53,039,571	5.63	T	T	C	0.08
	AX-90397199	3	43,985,440	6.00	C	C	T	0.08
	AX-90440743	13	21,445,098	6.08	C	C	T	0.08
	AX-90525380	20	2,448,280	6.48	A	A	T	0.10
Phe	AX-90525380	20	2,448,280	5.44	A	A	T	0.10
Pro	AX-90332294	1	6,633,674	6.53	G	T	G	0.07
	AX-90522787	3	43,983,721	7.58	G	G	T	0.08
	AX-90374011	19	43,104,938	5.39	A	A	G	0.27
Ser	AX-90332294	1	6,633,674	5.43	G	T	G	0.07
	AX-90522787	3	43,983,721	5.57	G	G	T	0.08
Thr	AX-90332294	1	6,633,674	5.50	G	T	G	0.07
	AX-90522787	3	43,983,721	6.26	G	G	T	0.08
Val	AX-90333532	6	4,923,700	5.10	G	A	G	0.15

SNP, single-nucleotide polymorphism; Chr, chromosome; Minor, minor allele; Major, major allele; MAF, minor allele frequency; + gene model: Glyma.Wm82.a2.v1.

**Table 5 plants-12-01665-t005:** Candidate genes associated with protein and oil content in wild soybean.

Trait	SNP	Chr	Position	Candidate Gene ^+^	Location (bp)	Gene Description
Protein	AX-90422214	11	1,067,467	*Glyma.11g015500*	1,066,054..1,068,916	RNA-binding (RRM/RBD/RNP motifs) family protein
	AX-90436656	12	4,681,230	*Glyma.12g063800*	4,680,402..4,682,246	OBF-binding protein 3
	AX-90336510	13	24,160,907	*Glyma.13g128700*	24,151,824..24,170,639	Myosin 2
	AX-90450715	14	44,061,946	*Glyma.14g179200*	44,060,802..44,063,722	10-formyltetrahydrofolate
	AX-90368184	15	4,332,019	*Glyma.15g055200*	4,330,953..4,332,402	F-box and associated interaction domain-containing protein
Oil	AX-90513548	12	34,460,746	*Glyma.12g183500*	34,455,733..34,463,207	Homolog of histone chaperone HIRA
	AX-90440743	13	21,445,098	*Glyma.13g099400*	21,442,358..21,445,246	Metaxin-related
	AX-90525501	14	8,179,883	*Glyma.14g089900*	8,175,666..8,180,225	SWIM zinc finger family protein
	AX-90387626	20	4,185,011	*Glyma.20g032400*	4,183,317..4,191,520	EXS (ERD1/XPR/SYG1) family protein
	AX-90339137	20	10,234,522	*Glyma.20g050300*	10,226,320..10,241,515	Zinc-binding alcohol dehydrogenase family protein
	AX-90513791	20	32,766,317	*Glyma.20g087700*	32,765,127..32,768,858	Protein of unknown function (DUF668)

SNP, single-nucleotide polymorphism; Chr, chromosome; + gene model: Glyma.Wm82.a2.v1.

**Table 6 plants-12-01665-t006:** Candidate genes associated with amino acid contents in wild soybean.

Trait	SNP	Chr	Position	Candidate Gene ^+^	Location (bp)	Gene Description
Ala, Asp, Glu, Gly, Leu, Lys, Pro, Ser, Thr	AX-90332294	1	6,633,674	*Glyma.01g053100*	6,620,086..6,625,502	Translation initiation factor 2C (eIF-2C) and related proteins
				*Glyma.01g053200*	6,632,524..6,638,087	Prefoldin chaperone subunit family protein
				*Glyma.01g053900*	6,835,875..6,836,372	Aconitase
Ala, Asp, Glu, Gly, Leu, Lys, Pro, Ser, Thr	AX-90522787	3	43,983,721	*Glyma.03g239700*	43,835,476..43,838,866	Aspartyl protease/7S seed globulin precursor
				*Glyma.03g240100*	43,884,764..43,895,501	D-amino acid aminotransferase-like PLP-dependent enzyme superfamily protein
				*Glyma.03g242400*	44,043,853..44,051,476	Shikimate dehydrogenase
Val	AX-90333532	6	4,923,700	*Glyma.06g061700*	4,644,103..4,646,096	Chorismate mutase
				*Glyma.06g064700*	4,921,259..4,925,737	Xanthine/uracil permease family protein
Arg, His	AX-90422214	11	1,067,467	*Glyma.11g014900*	1,029,017..1,031,439	Acyl-activating enzyme
				*Glyma.11g015500*	1,066,054..1,068,916	RNA-binding (RRM/RBD/RNP motifs) family
Ala	AX-90390655	12	5,348,412	*Glyma.12g072500*	5,344,610..5,350,276	Arogenate dehydratase 2
Lys	AX-90440743	13	21,445,098	*Glyma.13g099400*	21,442,358..21,445,246	Metaxin-related
Ala	AX-90434969	15	1,252,401	*Glyma.15g012300*	980,507..987,672	Aminotransferase
				*Glyma.15g015800*	1,250,483..1,253,447	
Pro	AX-90374011	19	43,104,938	*Glyma.19g164800*	42,559,312..42,561,641	Glycinin subunit G7
				*Glyma.19g164900*	42,567,390..42,570,310	Glycinin A1bB2-784
Arg, Lys, Phe	AX-90311210	20	2,947,656	*Glyma.20g025400*	2,768,770..2,781,380	Asparagine synthase

SNP, single-nucleotide polymorphism; Chr, chromosome; + gene model: Glyma.Wm82.a2.v1.

## Data Availability

The original contributions presented in the study are publicly available.

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
