# Peer review of "A Genome-Wide Association Study of Protein, Oil, and Amino Acid Content in Wild Soybean (Glycine soja)"

_plants, 2023, doi:10.3390/plants12081665_

Round 1

Reviewer 1 Report

The authors of this study used SNP to GWAS analysis to investigate the contents of proteins, amino acids, and oils. While the study's aims are clear and the authors identified several candidate genes based on GWAS results, a potential weakness is that they did not verify the expression of these genes using qPCR or other tools. Upon examining the candidate genes listed in Table 5, it was challenging to establish a clear link with the contents of proteins, amino acids, and oils. As such, I suggest that the authors either conduct qPCR or engage in further discussion to address this issue.

Regarding the significant SNPs identified in the study, the authors reported detecting a total of 44 SNPs in their abstract. However, in their description, they stated that "Based on a significance threshold of -log10(P) > 6.29, 13 SNPs across 6 chromosomes and 30 SNPs across 4 chromosomes were detected for protein and oil, respectively (Figure 4)." (LINE 139-140). The sum of 13 and 30 is 43, so I recommend that the authors verify all values in the manuscript.

Author Response

Thanks to the reviewer for the good advice

Reviewer 2 Report

The T      The authors conducted a large-scale study, the results of which will certainly be usef    be useful in soybean breeding. The manuscript may be accepted after consid     consideration by the authors of the comments below.

 The question remains how the content of specific amino acids is associated with certain protein-coding genes. The authors made some assumptions in this regard, but they seem unconvincing. It would be more honest to admit that this question remains open for the time being.

 The phrase on lines 285-87 “Because protein complexes form according to the arrangement and composition of their constituent amino acids, Glyma.01g053200 may be involved in changes to the contents of various amino acids” remained incomprehensible to me. The fact that 7S and 11S proteins make up a significant part of storage proteins is not associated with the presence of SNPs in their structural genes. The possible involvement of some genes in the regulation of amino acid biosynthesis is again difficult to associate with mutations in their structural genes, leading to a change in the content of amino acids in proteins.

 By default, the authors refer the quantities of amino acids they determine to the amino acids of proteins. Based on the literature data (doi: 10.1017/S1479262121000071), the content of free amino acids is about 1% of the amino acids in proteins, which cannot affect the conclusions drawn in the work. However, this should probably be mentioned in the article.

 At last, a technical note: correct the list of references according to the journal’s rule: “For documents co-authored by a large number of persons (more than 10 authors), you can either cite all authors, or cite the first ten authors, then add a semicolon and add ‘et al.’ at the end.”

Author Response

(The authors gave the same response as above.)

Round 2

Reviewer 2 Report

Thank you for correction.